# EXCAVATING CONSISTENCY ACROSS EDITING STEPS FOR EFFECTIVE MULTI-STEP IMAGE EDITING

## ABSTRACT

Multi-step image editing with diffusion models typically requires repeatedly executing the inversion–denoising paradigm, which leads to severe challenges in both image quality and computational efficiency. Repeated inversion introduces errors that accumulate across editing steps, degrading image quality, while regeneration of unchanged background regions incurs substantial computational overhead. In this paper, we present ExCave, a training-free multi-step editing framework that improves both image quality and computational efficiency by excavating consistency across editing steps. ExCave introduces an inversion sharing mechanism that performs inversion once and reuses its consistent features across subsequent edits, thereby significantly reducing errors. To eliminate redundant computation, we propose the CacheDiff method that regenerates only the edited regions while reusing consistent features from unchanged background regions. Finally, we design GPU-oriented optimizations to translate theoretical gains into practical reductions in end-to-end latency. Extensive experiments demonstrate that ExCave achieves superior image quality and dramatically reduces inference latency, establishing a new paradigm for accurate and efficient multi-step editing.

## 1 INTRODUCTION

With the breakthrough progress of diffusion models (Dalva et al., 2024; Esser et al., 2024; Peebles & Xie, 2023; Xie et al., 2024), they have become the state-of-the-art methods for image editing tasks (Cao et al., 2023; Hertz et al., 2022; Xu et al., 2023). Diffusion-based image editing methods are widely applied in various domains, including image inpainting (Hertz et al., 2022; Lu et al., 2023), image composition (Wang et al., 2024b; Xue et al., 2022), and image enhancement (Yi et al., 2023; Zhou et al., 2023). Such methods typically adopt the inversion–denoising paradigm (Mokady et al., 2023; Wang et al., 2024a): (1) the inversion stage maps the input image to the corresponding latent-space noisy image, and then (2) the denoising stage gradually removes noise and modifies the image structure to generate the edited image.

In practical image editing scenarios, users' preferences are highly individualized and often uncertain, making it challenging for a single-step editing process to consistently meet specific tastes. For instance, when adding a cute animal to an image, the single-step process might insert a cat. However, the user may find the cat insufficiently cute and request further modifications, such as increasing its fluffiness. Consequently, to achieve personalized image editing, users tend to iteratively refine prompts and perform multiple edits (defined as multi-step editing) until satisfactory results are obtained. To meet this demand (Joseph et al., 2024; Zhou et al., 2025), existing frameworks must repeatedly execute the inversion-denoising paradigm. However, this iterative mode poses severe challenges in both image quality and computational efficiency. Specifically, due to the discretization and causality of the inversion stage (Wang et al., 2024a; Zhu et al., 2025), frequent calls to inversion introduce substantial errors, progressively degrading image quality with increasing editing steps. Moreover, such an iterative mode incurs high computational overhead, posing severe challenges for real-time image editing. Consequently, traditional multi-step editing frameworks fail to achieve satisfactory image quality and generation speed, necessitating the design of an accurate and efficient multi-step editing framework.

To address this problem, we conduct an in-depth analysis of traditional multi-step editing frameworks (Joseph et al., 2024; Zhou et al., 2025) and identify that **their inefficiency stems from**

**their neglect of region consistency, where some image regions remain unchanged across editing steps**. Specifically, we find that across successive editing steps, only the regions semantically relevant to the edit prompts necessitate transformation, while irrelevant areas remain consistent. Fig. 1 illustrates the changes in different image regions during two-step editing. For example, in the first editing step, the user intends to add a hiking stick to the mountaineer. Consequently, the hand region, which is highly relevant to the hiking stick, is identified as the **edited region** (red box in Fig. 1) and is modified through the inversion–denoising paradigm. In contrast, regions unrelated to any prompts, such as the mountain area, are regarded as **background regions** (covered by a blue box) and remain unchanged.

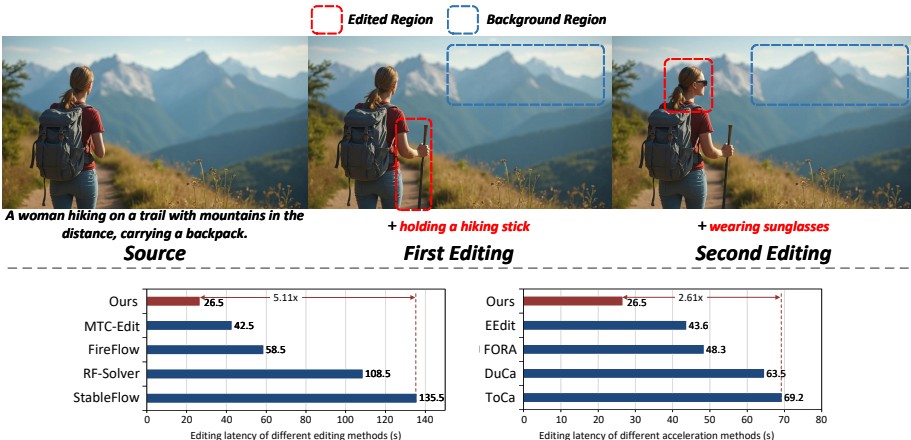

Figure 1: An example of ExCave in two-step editing and efficiency comparisons.

Motivated by these consistency properties, we propose a training-free multi-step editing framework, termed ExCave, achieving improvements in both image quality and computational efficiency. To exploit region consistency, we propose the inversion sharing mechanism. This mechanism initializes consistent and inconsistent features through the first inversion stage and then shares them across the subsequent editing steps. During each editing step, the inconsistent features corresponding to the edited regions are identified and updated by the denoising stage, ensuring that critical information is fully preserved. By requiring only one inversion stage regardless of the number of editing steps, our approach drastically reduces the accumulated errors introduced by repeated inversion. Furthermore, leveraging the acceleration potential provided by region consistency, we develop the CacheDiff method, which directly retrieves shared features corresponding to background regions without performing redundant computations on unmodified areas. For the edited regions, we adopt a sparse dataflow into the denoising stage to selectively regenerate only the necessary content, thereby avoiding excessive computation. Finally, to facilitate the deployment of ExCave, three GPU optimization techniques are introduced to convert the theoretical computational gains into actual reductions in end-to-end latency, thereby promoting its practical use.

In summary, our contributions are as follows: (1) We propose a training-free multi-step editing framework, ExCave, which leverages the region consistency of multi-step editing to achieve improvements in both image quality and computational efficiency. (2) We introduce the inversion sharing mechanism and the CacheDiff method to harness the consistency of regions, respectively improving the accuracy and computational efficiency of the multi-step editing. (3) We design three GPU optimization techniques to translate the computational benefits of ExCave into practical latency reduction, enhancing its usability. (4) Extensive experiments demonstrate that ExCave improves image quality in multi-step editing while reducing runtime latency.

## 2 RELATED WORK

### 2.1 IMAGE EDITING

As an important application in the field of image generation, image editing has been extensively studied and explored in academia. Common editing approaches follow the noise-addition and noise-reduction framework (i.e., the inversion-denoising paradigm), where the original image is progres-

sively perturbed by a certain level of noise in the latent space, and then the denoising capability of diffusion models is utilized to gradually obtain the final edited image.

When the inversion–denoising paradigm is applied to editing tasks such as prompt-guided editing (Wang et al., 2024a; Cao et al., 2023; Hertz et al., 2022), image composition (Wang et al., 2024b; Lu et al., 2023), and image dragging (Shi et al., 2024; Zhao et al., 2024), it usually involves operations on the attention features, including modification, enhancement, and replacement. For example, the RF-Solver (Wang et al., 2024a) model caches the $V$ matrices of the last few timesteps during the inversion stage, and in the denoising stage, it replaces the $V$ matrices generated at the corresponding timesteps with the cached ones for attention computation, thereby ensuring that the edited image retains similar to the original image.

## 2.2 DIFFUSION MODEL ACCELERATION

Low-latency and high-quality image generation is an important research field. Current diffusion model acceleration approaches mainly fall into two categories: reducing the number of sampling steps (Xue et al., 2023; Zheng et al., 2023; Gonzalez et al., 2023) and accelerating internal computations of diffusion models (Yan et al., 2025; Zou et al., 2025; Selvaraju et al., 2024; Zou et al., 2024). Since reducing the number of sampling steps significantly affects image quality, it is unsuitable for image editing. The mainstream solution for reducing internal computation is timestep-level token caching, which skips the computation of less important tokens in the current timestep by reusing tokens computed in previous timesteps. Unfortunately, existing caching schemes are designed for single-step editing and overlook optimization opportunities arising from the consistency properties of multi-step editing, making them inefficient in multi-step scenarios.

Moreover, existing token caching schemes suffer from the following issues: first, they require users to pre-mark the locations of edited regions, but editing tasks such as prompt-guided editing cannot obtain this information in advance, thus greatly limiting applicability. Second, these caching schemes are essentially variants of approximate computation, which inevitably degrade image quality. Additionally, they are not optimized for GPUs, so their improvements remain at the theoretical level and cannot be translated into practical end-to-end latency reduction.

Our proposed multi-step editing framework effectively reduces temporal and spatial redundancy by reusing features across editing steps, thereby significantly enhancing computational efficiency. In addition, we propose a series of GPU-oriented optimizations to transform the computational gains into practical end-to-end latency reduction, strengthening the practicality of our framework.

## 3 PRELIMINARIES

### 3.1 IMAGE EDITING PARADIGM

$$X_{t_i} = X_{t_{i-1}} + (t_i - t_{i-1}) \times M_\theta(C, X_{t_{i-1}}, t_{i-1}), i \in \{1, ..., N\} \tag{1}$$

Traditional image editing methods perform editing based on the inversion-denoising paradigm. Specifically, they first execute the inversion stage, taking the original image $X_{t_0}$ as input and gradually adding noise under the constraint of the initial condition $C$ to obtain the noisy image $X_{t_N}$. As shown in Eqn. 1, given discrete timesteps $t = \{t_0, \ldots, t_N\}$, the model $M_\theta$ predicts the noise $M_\theta(C, X_{t_{i-1}}, t_{i-1})$ at each timestep $t_i$ according to the constraint $C$, and adds it to the input image $X_{t_{i-1}}$ to obtain the output image $X_{t_i}$. Then, in the denoising stage, the model gradually removes noise from the noisy image $Z_{t_N}$ ($Z_{t_N} = X_{t_N}$) under the editing condition $C'$ to obtain the edited image $Z_{t_0}$, as detailed in Eqn. 2.

$$Z_{t_{i-1}} = Z_{t_i} + (t_{i-1} - t_i) \times M_\theta(C', Z_{t_i}, t_i), i \in \{N, ..., 1\} \tag{2}$$

The key to realizing the inversion–denoising paradigm is the diffusion model $M_\theta$, which models the probability flow path from the noise distribution to the real image distribution by learning the forward simulation system defined by the ordinary differential equation (ODE) $dZ_t = v(Z_t, t)dt$. Owing to the reversibility of the ODE, $M_\theta$ can also support the transformation from the real image distribution to the noise distribution. This property endows image editing methods based on the inversion–denoising paradigm with high flexibility, making them the mainstream approach.

## 3.2 Rethinking Traditional Image Editing Paradigm

Traditional multi-step editing frameworks require repeated execution of the full inversion-denoising paradigm, which leads to two major challenges: poor image quality and low computational efficiency. Here, we conduct a detailed analysis of the causes of these two issues.

We first analyze the cause of poor image quality. Theoretically, if the editing condition $C'$ is identical to the initial condition $C$, meaning that no edited regions exist, the image should remain unchanged after processing through the inversion and denoising stages.

$$Z = denoising(inversion(Z, C), C) \tag{3}$$

However, due to discretization and causality in the inversion stage, researchers have found that even in the absence of edited regions, the image cannot be perfectly restored after the inversion and denoising stages. This implies that the inversion stage introduces non-negligible errors. Moreover, as the number of editing steps increases, the errors introduced during the inversion stage progressively propagate and amplify, ultimately causing image quality to degrade.

Next, we explore the cause of low computational efficiency. Traditional methods regenerate all regions of the image during the denoising stage. Since the edited regions occupy only $14.7\%$ of the image on average in multi-step editing, regenerating the unchanged background introduces considerable unnecessary computation, severely reducing the efficiency of multi-step editing.

The above issues indicate that traditional multi-step editing frameworks not only introduce non-negligible errors but also perform excessive redundant computation, resulting in both poor image quality and low computational efficiency. Hence, it is imperative to develop a more accurate and efficient editing framework.

## 3.3 Exploring New Opportunities from Region Consistency

Since the aforementioned problems are closely related to the background regions, we conduct experiments to analyze their characteristics in depth. We performed multi-step editing on images and analyzed the intermediate features. The detailed procedures and results are provided in the Appendix D. The experimental results show that: (1) across different editing steps, the features corresponding to the background regions exhibit very high consistency between two successive inversion stages; and (2) within a single editing step, the background features produced by the inversion and denoising stages are also highly consistent.

Inspired by these findings, we propose a new design principle: reusing features of the background regions across different editing steps and within a single step. Specifically, we reuse the features of background regions from the first inversion stage in subsequent inversion stages, thereby effectively preventing error propagation and amplification during multi-step editing. Furthermore, during the denoising stage, we reuse the background features obtained from the inversion stage for the denoising stage, thus avoiding redundant regeneration of the background.

## 4 Methodology

Based on the design principle in Section 3.3, we propose a multi-step editing framework named ExCave, which incorporates a feature cache to store shared features. To address the problem of poor image quality, we introduce the inversion sharing mechanism (Section 4.1). This mechanism initializes the feature cache using the noisy image and background features generated in the first inversion stage, and then shared them across all editing steps. In this way, the accumulated error introduced by repeated inversion is greatly reduced. Since edited regions exist in the image, the inversion sharing mechanism updates the feature cache with the newly generated features corresponding to the edited regions at each step, ensuring that complete features are retained for the denoising stages. To tackle the issue of low computational efficiency, we propose the CacheDiff method (Section 4.2). Given that the background features have already been stored in the feature cache, during the denoising stage of each editing step, we only identify and regenerate edited regions while directly reusing the cached background features, which avoids redundant computation on background regions and significantly improves efficiency. Finally, three GPU optimization techniques (Section 4.3) are introduced to convert the computational gains of ExCave into actual end-to-end latency reduction, thereby facilitating its practical deployment. Fig. 2 illustrates the overall workflow of ExCave.

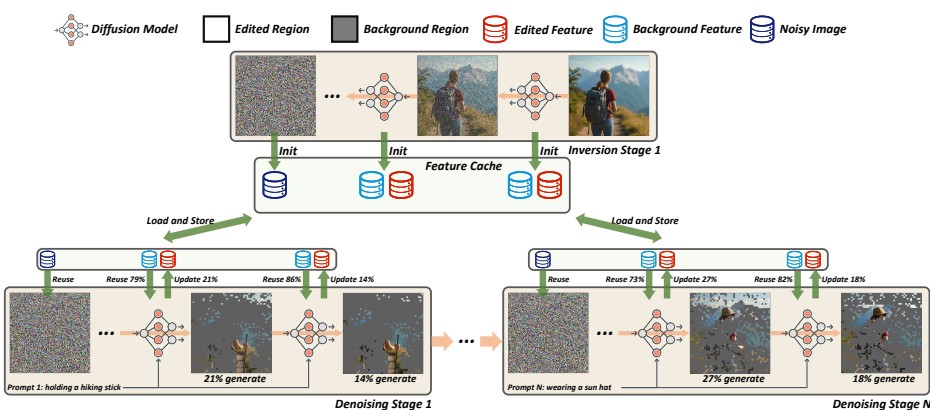

Figure 2: The overall workflow of the ExCave multi-step editing framework.

## 4.1 INVERSION SHARING MECHANISM (ISM)

The inversion sharing mechanism is designed with the purpose that all editing steps share the information generated in the first inversion stage. In this way, only a single inversion stage is required, which maximally reduces the impact of errors introduced by the inversion stage. As shown in Fig. 2, considering that the inversion stage provides both the noisy image and intermediate features for the denoising stage, we need to implement the sharing of these two components separately.

We first examine whether sharing the noisy image affects editing correctness. According to Eqn. 3, we find that the input noisy image $X_{t_N}^k$ in the denoising stage of the $k$-th editing step is identical to the noisy image produced by the inversion stage of the $(k+1)$-th step:

$$X_{t_N}^k = \text{inversion}(\text{denoising}(X_{t_N}^k, C^k), C^k) \qquad (4)$$

This indicates that each editing step produces the same noisy image in its inversion stage, i.e.,

$$\text{inversion}(Z_i, C_i) = \text{inversion}(Z_j, C_j), \quad (i \neq j) \qquad (5)$$

Therefore, we consider the noisy image obtained from a single inversion stage sufficient to be directly shared across all denoising stages. Based on this, in the inversion sharing mechanism, we first apply inversion stage to the original image once to obtain the noisy image, and then share it with each denoising stage, thereby enabling the sharing of noisy images across all editing steps.

The challenge of sharing features lies in the fact that, after each edit, the cached features corresponding to the edited regions become outdated, which would lead to incorrect results in subsequent steps. This implies that the feature cache must be updated during each editing. Given that, according to Eqn. 4, the features in the denoising stage of the $k$-th editing step is identical to that in the inversion stage of the $(k+1)$-th step, we introduce a neighboring update mechanism, which updates the feature cache using the features from the denoising stage of step $k$, thereby restoring the state of the inversion stage for step $k+1$. Specifically, during each denoising stage, the features consist of background features reused from the feature cache and newly generated features for the edited regions. Therefore, updating the features corresponding to the edited regions into the feature cache suffices to maintain the correct state of the inversion stage for the next editing step. Through this reuse-and-update strategy, we maximally realize the sharing of inversion features in multi-step editing. Notably, an essential prerequisite for updating cache is the accurate localization of the edited regions, which will be elaborated in detail in Section 4.2.1.

## 4.2 CACHEDIFF

Based on the region consistency, it can be inferred that the background regions remain unchanged during a single editing step, which implies that the background features in the inversion and denoising stages are identical. Accordingly, we propose the CacheDiff method, which reuses the background features from the inversion stage during denoising, thereby avoiding regeneration of the background. We first propose the Visual-Semantic Fusion (VS Fusion) localization method (Section 4.2.1) to accurately locate the edited regions in the denoising stage. Then, we design a reuse-based sparse dataflow for the denoising stage (Section 4.2.2), which generates only the edited regions while directly reusing the cached background features.

Fig. 2 illustrates the overall process of CacheDiff. Specifically, in the inversion stage, CacheDiff stores the input matrix of each timestep and the Key-Value pairs of each layer into the feature cache. During denoising, CacheDiff first locates the edited and background regions at the beginning of each timestep using VS Fusion. Then, it computes only the edited regions and directly reuses the background features from the feature cache. Notably, CacheDiff updates the feature cache with the Key-Value pairs of the edited regions, which is necessary to enable the neighboring update mechanism in Section 4.1.

### 4.2.1 VISUAL-SEMANTIC FUSION LOCALIZATION METHOD (VS FUSION)

---

**Algorithm 1:** VS Fusion Method

**Input:** Cross attention map $S \in R^{N \times M}$;
    Current image tokens $\{I_j^c\}_{j=0}^{N-1}$;
    Previous image tokens $\{I_j^p\}_{j=0}^{N-1}$;
    semantic threshold $P_s$;
    visual threshold $P_v$;
**Output:** localization mask $Mask$.
    // Generate semantic mask
1   $S_{avg} \leftarrow 0$;
2   **for** $i \leftarrow 0$ *to* $M$ **do**
3     $S_{avg} \leftarrow S_{avg} + S[:, i]$;
4   $S_{avg} \leftarrow S_{avg}/M$;
5   $M_s \leftarrow S_{avg} > P_s$;
    // Generate visual mask
6   **for** $i \leftarrow 0$ *to* $N$ **do**
7     $M_v[i] \leftarrow \sum_k |I_i^c[k] - I_i^p[k]|$;
8   $M_v \leftarrow M_v > P_v$;
9   $Mask \leftarrow M_s \wedge M_v$;
10   **return** $Mask$;

---

**Algorithm 2:** Reuse-Based Sparse Dataflow.

**Input:** image tokens $I$;
    localization mask $Mask$;
    previous key-value pairs $\{K^p, V^p\}$;
**Output:** edited image tokens $I^e$;
    updated key-value pairs $\{K^u, V^u\}$.
1   $I^e \leftarrow select(I, Mask)$;
    // sparse attention module
2   $\{Q^e, K^e, V^e\} \leftarrow qkv(I^e)$;
3   $\{K^b, V^b\} \leftarrow select(\{K^p, V^p\}, Mask)$;
4   $K \leftarrow concat(K^e, K^b, Mask)$;
5   $V \leftarrow concat(V^e, V^b, Mask)$;
6   $S^e \leftarrow Q^e \times K^T$;
7   $I_{tmp}^e \leftarrow Softmax(S^e) \times V$;
    // sparse MLP module
8   $I_{tmp}^e \leftarrow FC_1(I_{tmp}^e)$;
9   $I_{tmp}^e \leftarrow GELU(I_{tmp}^e)$;
10   $I^e \leftarrow FC_2(I_{tmp}^e)$;
11   **return** $I^e, K, V$;

---

To generate only the edited regions in the denoising stage, how to accurately locate them becomes a critical challenge. We observe that edited regions are determined by the editing prompts. Therefore, the relevance between image tokens and editing prompts can roughly indicate whether a token located within the edited regions, thereby enabling coarse-grained localization. Additionally, the difference between the input image of each denoising step and the corresponding image from the inversion stage can directly reflect the visual changes of each image region, which helps us determine the edited regions at a fine-grained level. Based on the above observations, we propose the Visual-Semantic Fusion Localization method, whose implementation is described in Alg. 1.

### 4.2.2 REUSE-BASED SPARSE DATAFLOW

After locating the edited regions, another key challenge is how to perform computation exclusively on these regions during the denoising stage. To address this, we first use the feature cache to store inversion-stage features for the reuse of background regions during denoising. We observe that the Key-Value pairs corresponding to the background contain complete feature information. Therefore, during inversion, CacheDiff stores the Key-Value pairs of each layer at each timestep in the feature cache for subsequent reuse according to the localization mask.

Then, we design a reuse-based sparse dataflow, as illustrated in Alg. 2. In the sparse attention module, CacheDiff loads the background key tokens $K^b$ and background value tokens $V^b$ from the feature cache (line3) and combines them with edited tokens to form the complete key matrix $K$ and value matrix $V$ (line4-5). Afterwards, $K$ and $V$ are combined with the edited query matrix $Q^e$ to perform the attention computation (line6-7). In a similar manner, the sparse MLP module is implemented to compute solely on the edited tokens.

### 4.3 GPU OPTIMIZATION TECHNIQUES

Through the inversion sharing mechanism and the CacheDiff method, ExCave improves image quality while significantly enhancing computational efficiency. However, when ExCave is integrated into baseline models, we find that the theoretical benefits are not fully translated into actual end-to-end latency reduction. Using Nsight System to profile GPU performance, we identify three major bottlenecks. The first issue is that allocating memory for Key-Value pairs stalls the GPU. Since we use CPU main memory as the feature cache, ExCave frequently calls CudaHostAlloc during the initial allocation of Key-Value pairs in the inversion stage, causing the GPU to be blocked and stay idle for

a long time. The second issue is that, in the default stream, accesses to the feature cache are executed serially with matrix multiplication operations. Consequently, the frequent cache accesses during the denoising stage significantly increase end-to-end latency. Finally, because ExCave accesses each Key-Value pair at most once during the denoising stage, this extremely low access frequency leads to frequent cache misses, further increasing latency.

**Pre-Allocation of Memory:** To avoid GPU stalls caused by initial memory allocations of Key-Value pairs, we propose pre-allocating memory technique. Specifically, leveraging the bidirectional communication capability of the PCIe bus, the GPU issues memory allocation requests to the CPU simultaneously as the model weights are loaded from CPU to GPU. In this way, we avoid memory allocation during the inversion stage, thereby improving GPU utilization.

**Multi-Stream Parallelism:** To mitigate the latency introduced by serial execution, we employ multi-stream parallelism to overlap cache accesses with GPU computations. We create three streams: a compute stream for model inference, a KV load stream for loading features of the background regions, and a KV store stream for writing back features of the edited regions. By running these three streams in parallel, most of the cache access latency is successfully hidden.

**Data Prefetching and Delayed Write-Back:** After deploying multi-stream parallelism, idle periods still exist in the compute stream, which arises from the cause that, within a single block, cache accesses and matrix operations have data dependencies (e.g., $K$ must be loaded before computing $Q \times K^T$), leaving insufficient room for data prefetching inside the block. To address this, we propose an inter-block data prefetching and delayed write-back technique, which decouples intra-block data dependencies through asynchronous data access, enabling full parallelism between cache access and matrix operations. Specifically, in block $i$, we prefetch $K_{i+1}^B$ and $V_{i+1}^B$ required by block $i + 1$, and pass $K_i^E$ and $V_i^E$ to block $i+1$ for write-back. Since block $i$ does not require $K_{i+1}^B, V_{i+1}^B$ and block $i + 1$ does not require $K_i^E, V_i^E$, this asynchronous data access successfully decouples dependencies, thereby enabling full parallelism between cache accesses and matrix computations.

## 5 EXPERIMENTS

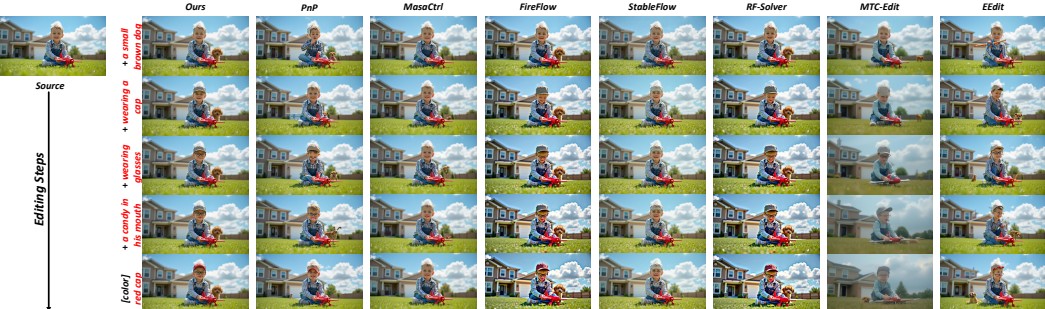

Figure 3: Qualitative comparison of multi-step editing results against baseline methods.

### 5.1 EXPERIMENTAL SETUP

**Baselines.** We compare our method with two categories of popular approaches: (1) Rectified Flow-based methods, including FireFlow (Deng et al., 2024), StableFlow (Avrahami et al., 2025), RF-Solver (Wang et al., 2024a), and MTC-Edit (Zhou et al., 2025); and (2) Diffusion-based methods, including PnP (Tumanyan et al., 2023) and MasaCtrl (Cao et al., 2023). In addition, we compare computational efficiency with the SOTA acceleration method EEdit (Yan et al., 2025), which skips the computation of less important tokens in the current timestep by reusing tokens computed in previous timesteps. In total, we evaluate seven widely adopted image editing methods, whose inference pipelines are built upon official implementations from Hugging Face or GitHub repositories.

**Datasets.** We adopt the PIE-Bench Benchmark (Ju et al., 2023) for image editing. Since the original PIE-Bench benchmark does not support multi-step image editing, we extend it to support five-step editing by following the prior work MTC-Edit (Zhou et al., 2025).

**Implementation Details.** Our method is implemented on FLUX-Dev (Labs, 2024), following the same framework as other Rectified Flow-based methods. We adopt the same hyperparameters as RF-Solver (Wang et al., 2024a), using 15 diffusion steps and setting the guidance values to 1 and 2 for inversion and denoising stages respectively. All experiments are conducted on an NVIDIA A100 GPU. More implementation details are provided in the Appendix.

**Metrics.** To comprehensively assess our approach, we adopt seven metrics spanning four evaluation dimensions. Overall image quality is evaluated using PSNR (Huynh-Thu & Ghanbari, 2008) and FID (Heusel et al., 2017), while background consistency is examined with LPIPS (Zhang et al., 2018), SSIM (Wang et al., 2004) and MSE. The CLIP-T score (Radford et al., 2021) is employed to measure text–image alignment. Finally, inference latency is reported to characterize computational efficiency.

## 5.2 EDITING RESULTS

**Qualitative Comparison.** We conduct extensive qualitative comparisons between our method and existing editing approaches, as shown in Fig. 3. Existing methods generate images that are visually similar to the sources but fail to preserve background consistency. In contrast, our framework shares the same background features across multiple editing steps, thereby effectively maintaining consistency in the background. Moreover, as the editing steps increase, existing methods struggle to maintain text–image alignment. For instance, the methods aiming at improving the efficiency of multi-step editing, such as EEdit, show a sharp decline in fidelity after three editing steps. By mitigating errors introduced during the inversion stage, the inversion sharing mechanism in our framework ensures better text–image alignment and higher image quality, thereby supporting more editing steps while maintaining editing quality.

Table 1: Comparison of image quality across various methods in multi-step editing.

| Method | Overall Quality | | Background Consistency | | | Text Alignment |
|---|---|---|---|---|---|---|
| | PSNR ↑ | FID ↓ | LPIPS $_{\times 10^{-2}}$ ↓ | SSIM $_{\times 10^{-2}}$ ↑ | MSE $_{\times 10^{-2}}$ ↓ | CLIP-T ↑ |
| PnP | 7.23 | 77.24 | 85.03 | 26.47 | 18.39 | 20.59 |
| MasaCtrl | 7.43 | 78.09 | 86.18 | 25.75 | 19.21 | 20.09 |
| FireFlow | 8.35 | 45.25 | 84.62 | 27.65 | **16.52** | 21.19 |
| StableFlow | 8.34 | 51.09 | 86.67 | 28.03 | 17.81 | 21.31 |
| RF-Solver | 8.22 | 49.33 | 83.91 | 27.15 | 16.92 | 21.29 |
| MTC-Edit | **8.55** | 43.79 | 84.34 | 28.24 | 16.87 | 21.33 |
| EEdit | 7.96 | 66.96 | 90.34 | 24.15 | 17.41 | 21.09 |
| Ours | 8.39 | **40.43** | **82.41** | **28.38** | 16.84 | **21.33** |

**Quantitative Comparison.** As shown in Table 1, our method achieves significantly better performance on background consistency metrics such as LPIPS, SSIM, since it allocates computation only to the edited regions and reuses cached background feature. Furthermore, in terms of text alignment, our method achieves higher CLIP-T score than existing approaches, since our CacheDiff method accurately identifies prompt-relevant regions and prioritizes their feature computation. We also achieve competitive results on PSNR and FID, as the proposed inversion sharing mechanism effectively inhibits error propagation during the diffusion process, thereby preserving image quality.

Table 2: End-to-end inference latency ($s$) across various methods for multi-step editing.

| Method | Two Steps | | | Three Steps | | | Four Steps | | | Five Steps | | |
|---|---|---|---|---|---|---|---|---|---|---|---|---|
| | Inversion | Denoising | Total | Inversion | Denoising | Total | Inversion | Denoising | Total | Inversion | Denoising | Total |
| PnP | 342.82 | 214.08 | 556.91 | 514.23 | 321.12 | 835.35 | 685.64 | 428.16 | 1113.81 | 857.05 | 535.19 | 1392.25 |
| MasaCtrl | **5.61** | 10.64 | 16.25 | **8.42** | 15.91 | 24.33 | 11.27 | 21.24 | 32.51 | 14.03 | 26.51 | 40.54 |
| FireFlow | 11.82 | 11.65 | 23.47 | 17.74 | 17.48 | 35.22 | 23.62 | 23.24 | 46.86 | 29.52 | 29.09 | 58.61 |
| StableFlow | 9.61 | 45.46 | 55.07 | 14.43 | 68.15 | 82.58 | 19.27 | 90.82 | 110.09 | 24.06 | 113.51 | 137.57 |
| RF-Solver | 21.83 | 21.64 | 43.47 | 32.75 | 32.41 | 65.16 | 43.62 | 43.24 | 86.86 | 54.57 | 54.01 | 108.58 |
| MTC-Edit | 7.85 | 7.61 | **15.46** | 11.72 | 11.47 | 23.19 | 15.63 | 15.23 | 30.86 | 19.56 | 19.01 | 42.57 |
| EEdit | 9.79 | 7.63 | 17.42 | 14.72 | 12.05 | 26.77 | 19.36 | 15.21 | 34.57 | 24.44 | 19.12 | 43.56 |
| Ours | 10.91 | **7.14** | 18.05 | 10.93 | **9.52** | **20.45** | 10.92 | 12.31 | **23.23** | 10.94 | 15.38 | **26.62** |

**Computational Efficiency.** Table 2 reports a comparison of end-to-end inference latency. By generating only the edited regions and directly reusing background features across editing steps, the CacheDiff method substantially reduces the computational overhead of multi-step editing. Consequently, our method averagely achieves an $65.8\%$ reduction in end-to-end latency compared with conventional methods, demonstrating the efficiency of ExCave. In addition, compared with EEdit,

ExCave can achieve greater latency reduction as the number of editing step increases. This is because EEdit optimizes only within single-step editing and neglects the opportunities provided by consistency across editing steps, resulting in suboptimal speedup. Moreover, ExCave delivers higher quality than EEdit, as verified by Fig. 3.

### 5.3 ABLATION STUDY

We conduct extensive ablation studies to analyze the contributions of the inversion sharing mechanism and CacheDiff method to editing quality and computational efficiency.

| Method | Overall Quality | | Background Consistency | | | Text Alignment |
|---|---|---|---|---|---|---|
| | PSNR ↑ | FID ↓ | LPIPS $_{\times 10^{-2}}$ ↓ | SSIM $_{\times 10^{-2}}$ ↑ | MSE $_{\times 10^{-2}}$ ↓ | CLIP-T ↑ |
| Baseline | 8.32 | 53.33 | 80.91 | 27.15 | 16.22 | 21.29 |
| Ours w/o CacheDiff | 8.37 | 50.73 | 80.70 | 29.38 | 14.84 | 21.39 |

Table 3: Ablation study on inversion sharing mechanism.

**Ablation of the Inversion Sharing Mechanism (ISM).** As illustrated in Fig. 4, incorporating the inversion sharing mechanism substantially improves background consistency. Specifically, our method perfectly preserves the appearance of the dog, whereas the baseline method introduces white spots on its eyebrows, demonstrating that our method provides a superior visual experience. This improvement stems from the inversion sharing mechanism, which enables the background regions of the original image to be shared across editing steps, thereby preventing error propagation from affecting these regions. Quantitative results in Table 3 further support this observation, showing consistent improvements across multiple metrics, thereby confirming that the inversion sharing mechanism effectively reduces errors during editing.

**Ablation of the CacheDiff method.** Table 4 presents the changes in inference latency with and without the CacheDiff method. By skipping redundant regeneration of background regions, CacheDiff performs computation only on 27% of pixels, leading to a latency reduction of 70.9% in the denoising stage. This demonstrates that CacheDiff significantly enhances computational efficiency in multi-step editing.

| Method | Inference Latency | | |
|---|---|---|---|
| | Inversion | Denoising | Total |
| Baseline | 54.57 | 54.01 | 108.58 |
| Ours w/o ISM | 54.13 | 15.72 | 69.84 |

Table 4: Ablation study on CacheDiff.

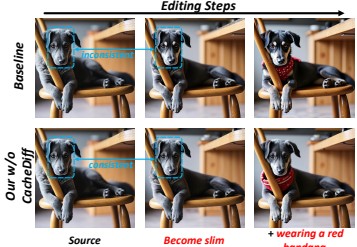

Figure 4: The qualitative ablation study about inversion sharing mechanism.

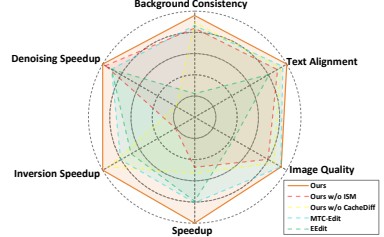

Figure 5: A multi-dimensional Comparison over different configurations and methods.

Fig. 5 presents a comprehensive comparison between our method and others across multiple dimensions. Our method achieves consistent advantages in image quality, background consistency, text alignment, and speedup, thereby validating its effectiveness.

### 6 CONCLUSION

We present ExCave, a framework that leverages region consistency to improve both the precision and efficiency of multi-step editing. By introducing the inversion sharing mechanism and the CacheDiff method, our framework suppresses error accumulation and avoids redundant computation. Experiments show that it achieves higher image quality and faster editing than existing approaches, demonstrating its practical value.

ETHICS STATEMENT

Our work focuses on optimizing multi-step image editing and thus does not have direct ethical implications. However, the capabilities of image generation and editing should be carefully considered to prevent misuse for producing harmful content such as gore or violence.

REPRODUCIBILITY STATEMENT

We provide additional experiments and implementation details in Appendix A, C, D, and E, including further experimental results and the proof of Section 3.3. The source code is available at `https://anonymous.4open.science/r/ExCave-D623`.

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

APPENDIX

## A  IMPLEMENTATION DETAILS

Experiments were conducted on a machine with the following hardware and software specifications:

### A.1  HARDWARE SPECIFICATIONS

- Architecture: x86 64
- CPU Op-Modes: 32-bit, 64-bit
- Address Sizes: 48 bits physical, 48 bits virtual
- Byte Order: Little Endian
- Total CPU cores: 80
- On-line CPU(s) List: 0–79
- Vendor ID: AuthenticAMD
- Model Name: AMD EPYC 7443 24-Core Processor
- CPU Family: 25

### A.2  SOFTWARE SPECIFICATIONS

- Operating System: Ubuntu 22.04.3 LTS
- CUDA: 11.8
- Python: 3.10.16
- huggingface-hub: 0.31.2
- numpy: 2.2.5
- torch: 2.4.1
- transformers: 4.51.3

# B METRICS

Our experiments adopt a set of widely used metrics for image quality, prompt adherence, and efficiency. Frechet Inception Distance (FID) and Learned Perceptual Image Patch Similarity (LPIPS) are feature-based similarity metrics, which are computed using pretrained neural networks. Lower values indicate higher similarity. We use InceptionV3 for FID and AlexNet for LPIPS. Peak Signal-to-Noise Ratio (PSNR), Structural Similarity Index (SSIM), and Mean Squared Error (MSE) are pixel-space similarity metrics. Higher PSNR and SSIM, and lower MSE, indicate higher similarity. CLIP-T measures the alignment of generated images with the input prompts using a pretrained CLIP model. Higher scores indicate stronger adherence. In our experiments, we use the clip-vit-base-patch16 model. Inference latency quantifies the runtime overhead associated with model inference. Higher values indicate greater computational cost.

# C MORE EXPERIMENTS

In this section, we present more quantitative results, demonstrating the effectiveness of our method for multi-step editing in terms of both editability and structural preservation.

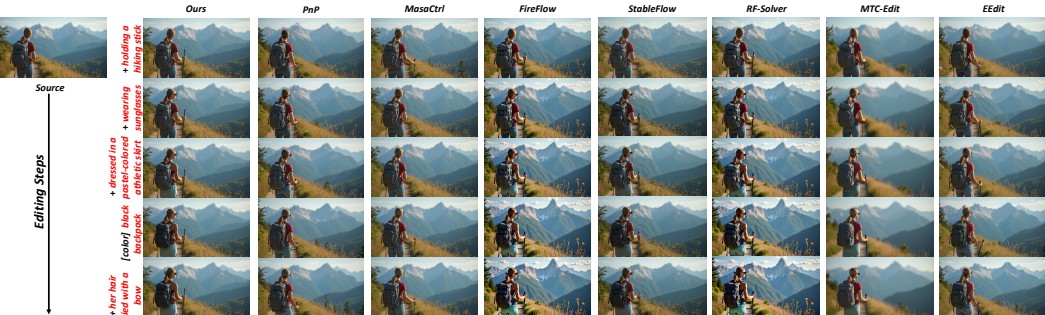

Figure 6: More qualitative comparison of multi-step editing results against baseline methods.

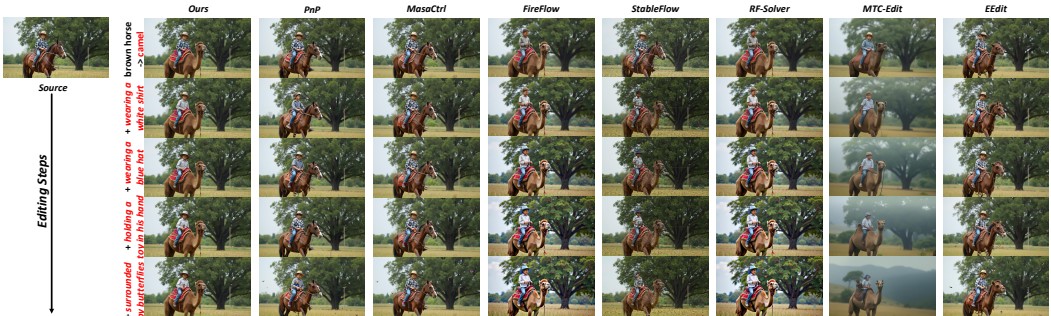

Figure 7: More qualitative comparison of multi-step editing results against baseline methods.

# D EXPERIMENTAL SUPPORT FOR SECTION 3.3

We design experiments to conduct an in-depth analysis of the characteristics of regional consistency. Specifically, we perform multi-step editing on the input images using the baseline model RF-Solver and save its intermediate features for analysis. Afterwards, we compare the feature similarity between the inversion and denoising stages within the same editing step, as well as between the inversion stages of adjacent editing steps. We then visualize these statistics across multiple timesteps using heatmaps to reveal general patterns.

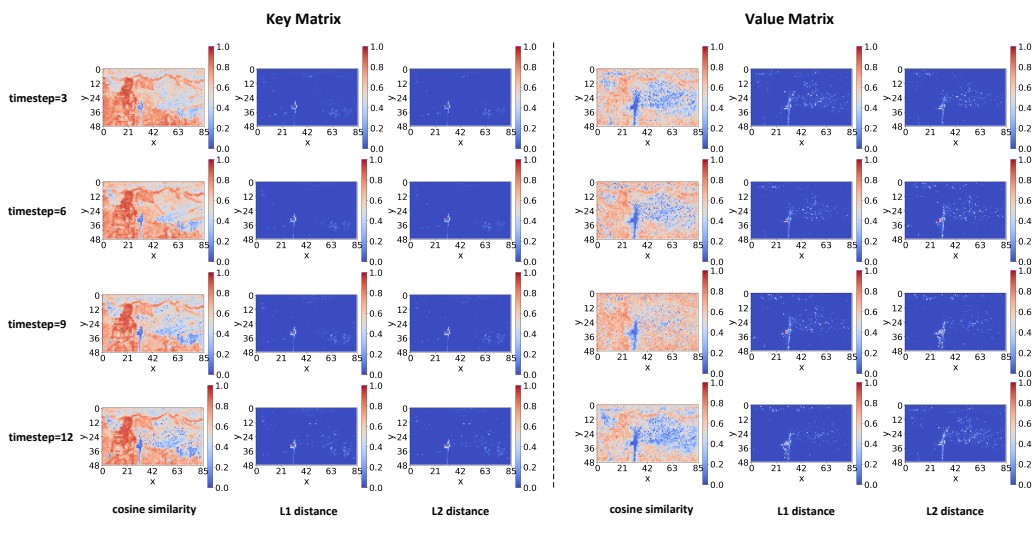

Figure 8: Token-level similarity of key matrix and value matrix between inversion stage and denoising stage.

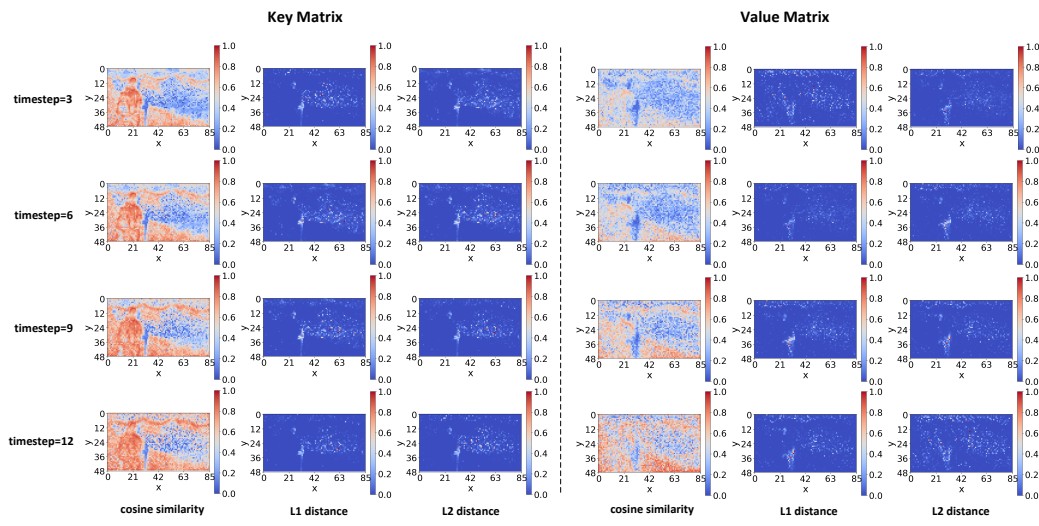

Figure 9: Token-level similarity of key matrix and value matrix between different inversion stages.

The results are shown in Fig. 8 and Fig. 9, where higher cosine similarity (deeper red) and lower L1/L2 distance (deeper blue) are better, indicating higher similarity. We find that intermediate features corresponding to background regions (i.e., consistency regions) exhibit consistently high similarity between the inversion and denoising stages within the same editing step, suggesting that these features can be safely reused across these two stages. Similarly, as illustrated in Fig. 9, background features also demonstrate high similarity across inversion stages of different editing steps, indicating that feature sharing is also feasible across multiple inversion stages.

## E    TRENDS OF IMAGE QUALITY METRICS WITH INCREASING EDITING STEPS

Fig. 10 presents the changes of PSNR across different editing steps. It can be find that the PSNR degradation of our method remains relatively small as the number of editing steps increases, consis-

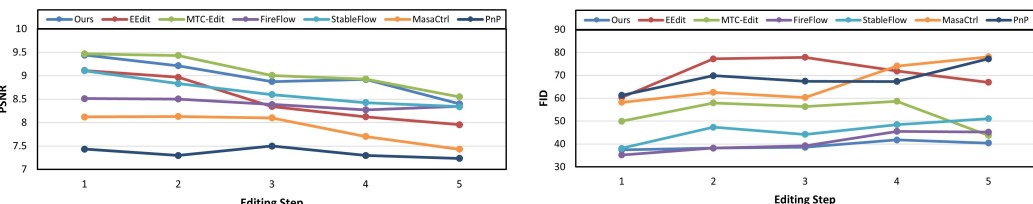

Figure 10: PSNR across different editing steps.     Figure 11: FID across different editing steps.

tently ranking among the Top-2 methods. These results clearly demonstrate the competitiveness of ExCave.

Fig. 11 illustrates the trend of FID with respect to editing steps. We observe that the FID growth of our method is the slowest, maintaining Top-1 performance in most cases. This advantage arises from our inversion sharing mechanism, which effectively suppresses error accumulation across multiple editing steps and thereby preserves higher image quality.

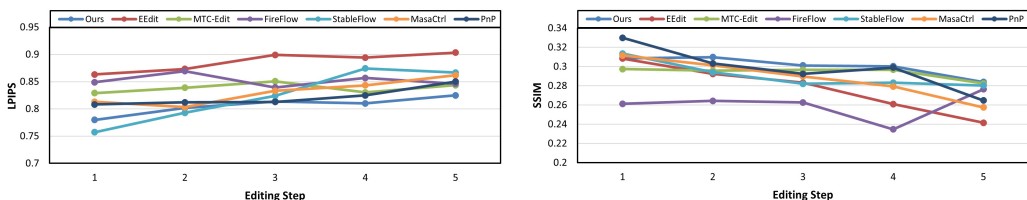

Figure 12: LPIPS across different editing steps.     Figure 13: SSIM across different editing steps.

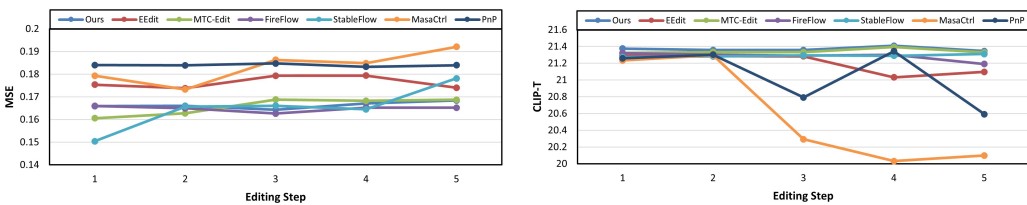

Figure 14: MSE across different editing steps.     Figure 15: CLIP-T across different editing steps.

Fig. 12-Fig. 14 report the changes of LPIPS, SSIM, and MSE. Our method consistently ranks among the best across all editing steps and its superiority becomes more pronounced as the number of editing steps increases. This can be attributed to our effective exploitation of regional consistency across multiple editing steps, which enables the sharing of background features from the original image and thus achieves superior background consistency.

Finally, Fig. 15 shows the trend of CLIP-T scores with respect to editing steps. Our method achieves consistently stable and leading CLIP-T performance, which can be explained by the proposed VS Fusion method. By accurately identifying regions relevant to the editing prompts and focusing generation on those areas, our method attains improved text–image alignment.

## F   THE USE OF LLMS

We use the large language models (LLM) to polish the writing. Specifically, we first write the entire paper independently, and then employed LLMs to refine sentences that are informal or insufficiently academic. LLMs are primarily used in Sections 3 and 4. For example, in the last paragraph of Section 3.2, we write: *Hence, it is imperative to develop a more accurate and efficient editing framework.* The initial version is *Therefore, it is necessary to design a more accurate and efficient editing*

*framework*. We consider the initial version do not adequately convey a sense of urgency, so the LLM refinement replace *necessary* with *imperative*. Moreover, since *Therefore* appears frequently in the original text, we adopt the LLM's suggestion to substitute it with *Hence*.

In summary, we mainly use LLMs to adjust sentence structures and word choices, without allowing them to modify the paper's content. Importantly, we do not use LLMs to retrieve references or generate research ideas, since hallucinations can lead to incorrect references or unreliable suggestions. Moreover, we regard idea generation as the core of our work, which must be carried out independently by authors.

