# OpenReview forum: "Excavating Consistency Across Editing Steps for Effective Multi-Step Image Editing"
_ICLR.cc/2026/Conference — ICLR 2026 Conference Withdrawn Submission_

### Official Review · Reviewer_t2qe · 2025-10-29

**Soundness:** 3
**Presentation:** 2
**Contribution:** 2
**Rating:** 4
**Confidence:** 4

**Summary:**

This paper tackles the challenges of inversion-based multi-turn image editing, where repeated inversions cause error accumulation and redundant computation. To mitigate these issues, the authors propose performing inversion only once and reusing intermediate features across multiple editing steps to preserve consistency and reduce degradation. To further enhance efficiency, they introduce CacheDiff, a caching mechanism that stores inversion results (from the input image to the noise space) for reuse in subsequent edits. CacheDiff selectively regenerates edited regions using localized cross-attention masks while keeping unedited background areas intact. Additionally, the authors implement GPU-oriented optimizations to accelerate their ExCave framework for practical, real-time multi-turn editing.

**Strengths:**

1. Overall assessment: The proposed method is straightforward and well-motivated.
2. Strength: The key advantage of the ExCave framework lies in its ability to selectively reuse unchanged features during multi-turn editing based on different editing prompts, thereby reducing redundant inversions and mitigating error accumulation across editing steps.

**Weaknesses:**

1. Ambiguity in terminology and writing: The paper’s writing is sometimes ambiguous. The term “Multi-Step Editing” may be misleading, as it could be interpreted as multiple denoising steps during diffusion inference. The authors are encouraged to adopt the term “multi-turn”, consistent with prior work such as “Multi-turn Consistent Image Editing” [1]. Additionally, the exposition in Section 4.1 is difficult to follow—particularly Equation (5), where the meanings of i and j are unclear.
2. Insufficient description of CacheDiff: The proposed CacheDiff method in Section 4.2 requires more detailed explanation, especially regarding the VS Fusion design in Algorithm 1 and the sparse attention module in Algorithm 2. This section also appears to lack sufficient experimental evidence to substantiate the claimed efficiency and accuracy improvements.
3. Potential limitation in generality: The proposed CacheDiff approach may be constrained to object-centric editing, as it depends on masks derived from VS Fusion (based on cross-attention maps). In more complex scenarios such as style or appearance editing, where both content and texture undergo substantial changes, reusing cached key–value features may not yield effective results.

[1] Zhou Z, Deng Y, He X, Dong W, Tang F. Multi-turn Consistent Image Editing. arXiv preprint arXiv:2505.04320. 2025 May 7.
[2] Zhang K, Mo L, Chen W, Sun H, Su Y. Magicbrush: A manually annotated dataset for instruction-guided image editing. Advances in Neural Information Processing Systems. 2023 Dec 15;36:31428-49.

**Questions:**

1. Lack of discussion on related work: The Related Work section omits prior studies on multi-turn editing, despite referencing “Multi-turn Consistent Image Editing” [1] in the experiments, which tackles the same problem as this paper. The authors should include a discussion of this and other related approaches to better contextualize their contributions.
2. Overly complex averaging operation in Algorithm 1: The averaging step in Algorithm 1 (Section 4.2.1, lines 1–4) appears unnecessarily complicated and could likely be simplified. Moreover, in line 7, the variable k (“turn k”) seems redundant, as the algorithm is executed within a single-turn editing process. Simplifying these formulations would improve clarity and readability.
3. Lack of motivation and citations for proposed methods: In Section 4, all the described components appear to be the authors’ own proposals, yet no references or motivations from prior work are provided. If these methods are indeed novel, additional experimental evidence or ablation studies are needed to substantiate their effectiveness.
4. Use of inappropriate evaluation benchmark: The experiments use the PIE dataset, which is not a benchmark for multi-turn editing. A more suitable dataset, such as MagicBrush [2], supports multi-turn evaluation and would provide a more convincing validation. Additionally, results should be reported for each editing turn to demonstrate how error accumulation evolves and how the proposed method compares against baselines across turns.
5. Fairness of comparison in Table 1: The strong visual quality in Table 1 may partly stem from using FLUX as the underlying diffusion backbone. Since the proposed method is designed to work across multiple diffusion models (e.g., SD and FLUX), fair comparisons should be performed under the same backbone used by competing methods.
6. Choice of attention maps in VS Fusion: In VS Fusion, why were cross-attention maps selected for mask extraction? Have the authors explored alternatives such as self-attention maps? An ablation comparing these choices would clarify the design rationale.
7. Motivation for preserving background via KV features: In Section 4.2.2, the paper states that KV features are preserved to maintain background information. What motivated this specific design choice? Could other representations (e.g., latent feature blending) be equally effective?
8. Clarification of “sparse attention” and “sparse MLP” modules: The “sparse attention” module in Algorithm 2 appears to be standard full attention with modified K and V matrices that mix edited and background tokens. The authors should clarify how this differs from standard attention and provide details on the “sparse MLP” component, ideally with diagrams or pseudocode.
9. GPU optimization clarity: In Section 4.3, the GPU-oriented optimizations are described at a high level. Including pseudocode or an illustrative figure would greatly enhance clarity and reproducibility.

---

### Official Review · Reviewer_HAzD · 2025-10-31

**Soundness:** 2
**Presentation:** 3
**Contribution:** 2
**Rating:** 4
**Confidence:** 4

**Summary:**

This paper proposes ExCave, a training-free framework for efficient and accurate multi-step image editing with diffusion models. Traditional approaches repeatedly perform inversion and denoising at each editing step, which causes accumulated errors and high computational cost. ExCave addresses these problems through two main ideas: inversion sharing, which performs inversion only once and reuses consistent features across edits, and CacheDiff, which regenerates only the modified regions while reusing unchanged background features. The authors also introduce GPU optimizations to further reduce latency. Experiments show that ExCave improves image quality and significantly speeds up inference, establishing a new paradigm for multi-step diffusion-based image editing.

**Strengths:**

Strengths:
1. The framework reuses inversion features across editing steps, which reduces error accumulation and improves image consistency.
2. It regenerates only the edited regions while caching unchanged areas, which significantly enhances computational efficiency.
3. It works without additional model training and includes GPU-oriented optimizations, making it practical for real-world multi-step editing.

**Weaknesses:**

Weaknesses
1. The method relies on the assumption that shared inversion features remain valid across large semantic edits, which may fail for major content changes.
2. The CacheDiff mechanism depends on accurate region masks, and errors at boundaries can cause visual artifacts or inconsistencies.
3. The paper does not compare against recent strong directions such as inversion-free editing methods and prior cache-based or region-reuse acceleration approaches, so generality and relative gains are unclear.

**Questions:**

1. How robust is the inversion sharing mechanism when the edits involve large semantic or structural changes? Would a partial re-inversion strategy help maintain consistency in such cases?
2. How sensitive is the method to inaccuracies in region masks or gradual region drift over multiple edits?
3. Could the authors provide quantitative or qualitative comparisons with inversion-free and other cache-based acceleration methods to better show where ExCave stands relative to these recent baselines?

---

### Official Review · Reviewer_jyn3 · 2025-10-31

**Soundness:** 3
**Presentation:** 3
**Contribution:** 3
**Rating:** 6
**Confidence:** 4

**Summary:**

This paper proposes ExCave, a training-free framework for multi-step image editing with diffusion models that enhances both image quality and computational efficiency. ExCave introduces two main innovations: (1) Inversion Sharing Mechanism (ISM): Performs inversion once and reuses consistent latent features across subsequent editing steps, reducing error propagation. (2) CacheDiff: A feature caching method that regenerates only edited regions while reusing background features, significantly reducing latency. Additionally, the authors introduce GPU-oriented optimizations to convert theoretical gains into real speedups. Experiments on an extended PIE-Bench dataset demonstrate consistent improvements in background consistency, text–image alignment, and latency.

**Strengths:**

- Novel observation: Recognizing region-level consistency across editing steps and leveraging it effectively.
- Practical implementation: GPU optimizations demonstrate attention to real-world deployment.

**Weaknesses:**

- No user study: While objective metrics are solid, perceptual or human preference evaluations would strengthen claims about “editing quality.”
- Comparison to state-of-the-art editing methods: The paper lacks comparisons to state-of-the-art image editing methods such as GPT-4o, Gemini2.5 (nano-banana).

**Questions:**

- How sensitive is ExCave to inaccuracies in the VS Fusion localization? Would misidentified edited regions harm image coherence?
- Can ExCave be combined with fine-tuned or training-based acceleration methods (e.g., ControlNet, LoRA) without retraining?

---

### Official Review · Reviewer_15y7 · 2025-10-31

**Soundness:** 3
**Presentation:** 2
**Contribution:** 2
**Rating:** 4
**Confidence:** 4

**Summary:**

The paper presents ExCave, a training-free framework for multi-step diffusion-based image editing that boosts both quality and efficiency. It targets two issues in iterative pipelines: error accumulation from repeated inversion and redundant computation on unchanged backgrounds. ExCave combines (1) an Inversion Sharing Mechanism (ISM) that performs inversion once and reuses consistent features across edits, and (2) CacheDiff, which sparsely recomputes only edited regions while reusing cached background features. With GPU optimizations (memory pre-allocation, multi-stream parallelism, and prefetching with delayed write-back), ExCave delivers up to 65.8% lower latency on extended PIE-Bench while maintaining or improving image quality and text–image alignment.

**Strengths:**

1. The paper tackles a practical and underexplored challenge in multi-step image editing, clearly diagnosing error accumulation from repeated inversion and redundant background regeneration, and proposes an edit-only keep-the-rest principle through inversion sharing and region-aware caching in a training-free manner.

2. The motivation-to-method logic is clear, moving from region-consistency observations to a concrete design that suppresses inversion errors and enables sparse recomputation, with sufficient technical detail and illustrative figures and pseudocode that make the pipeline easy to follow.

3. The GPU-oriented optimizations, including memory pre-allocation, multi-stream parallelism, and prefetching with delayed write-back, show strong attention to deployment by bridging algorithmic ideas with system-level efficiency and yielding substantial end-to-end latency reductions without sacrificing quality.

**Weaknesses:**

1. The paper states that edited regions are localized by comparing the denoising input with the corresponding inversion image, which “directly reflects visual changes,” but it is unclear precisely where in the pipeline this comparison is applied and how it interacts with the cache-consistency assumption that backgrounds remain reusable.

2. The VS Fusion locator depends on semantic and visual thresholds (Ps and Pv), yet the paper provides no systematic analysis of their sensitivity to prompt phrasing, illumination changes, or scale variations, nor an assessment of how false positives and false negatives propagate to image quality and end-to-end latency.

3. The GPU optimization section, while technically sound, appears tightly coupled to a specific implementation stack, raising concerns about portability and reproducibility.

4. The experimental setup emphasizes a single sampling configuration and solver choice on a limited set of backbones, which constrains the external validity of the conclusions. Additional experiments spanning step counts, solvers, and diverse backbones would better establish robustness and generalization.

**Questions:**

1. How robust is the VS Fusion–based localization to imprecise or compositional prompts, and what failure modes (if any) produce visible artifacts or degrade efficiency in multi-step editing?

2. Can the authors provide a systematic sensitivity study of the semantic and visual thresholds (Ps, Pv), including how false positives and false negatives affect image quality and end-to-end latency?

3. Have the authors evaluated the trade-off between cache size and latency gains under different GPU memory configurations, and how do these settings influence portability across hardware?

---

### Note · Authors · 2025-11-26

**Comment:**

After careful consideration, we have decided to withdraw our manuscript. We sincerely appreciate your efforts and the constructive feedback you provided. We will subsequently revise and improve the paper in accordance with your review comments.

**Withdrawal Confirmation:**

I have read and agree with the venue's withdrawal policy on behalf of myself and my co-authors.